# Nested PCR Detection of *Pythium* sp. from Formalin-Fixed, Paraffin-Embedded Canine Tissue Sections

**DOI:** 10.3390/vetsci9080444

**Published:** 2022-08-19

**Authors:** Nelly O. Elshafie, Jessica Hanlon, Mays Malkawi, Ekramy E. Sayedahmed, Lynn F. Guptill, Yava L. Jones-Hall, Andrea P. Santos

**Affiliations:** 1Department of Comparative Pathobiology, College of Veterinary Medicine, Purdue University, West Lafayette, IN 47907, USA; 2Department of Veterinary Clinical Sciences, College of Veterinary Medicine, Purdue University, West Lafayette, IN 47907, USA

**Keywords:** diagnosis, fungi, *Pythium insidiosum*, dog, biopsy, PCR, FFPE

## Abstract

**Simple Summary:**

*Pythium insidiosum* is a waterborne fungus-like organism commonly present in tropical and subtropical areas that causes a disease named pythiosis in dogs and other animals. This disease can cause inflammatory lesions on the skin and gastrointestinal tract and, if left untreated, may be fatal. Because this mold lacks the distinctive features of true fungi, it requires specific treatment. Diagnosis of pythiosis is commonly achieved by microscope visualization after staining the tissue sections (histopathology). Although some stains highlight hyphae, these techniques are challenging to distinguish *P. insidiosum* from other fungi. Our study aimed to develop a molecular technique (nested PCR) to specifically detect *P. insidiosum* DNA in biopsy specimens to aid in diagnosing this organism. Archived biopsies from 26 dogs suspected of pythiosis were examined by histopathology with special stains and tested by the novel nested PCR. Agreement between histopathology and nested PCR occurred in 18/26 cases. The microscopic examination identified hyphae consistent with *Pythium* sp. in 57.7% of the samples, whereas the nested PCR detected *P. insidiosum* DNA in 76.9% of samples, aiding in the sensitivity of the diagnosis of pythiosis in dogs. Using this combination of techniques, we report 20 canine cases of pythiosis over 18 years in Indiana and Kentucky, an unexpectedly high incidence for temperate climatic regions. Thus, we recommend using our nested PCR test in addition to the microscopic examination to increase the sensitivity of the diagnosis.

**Abstract:**

*Pythium insidiosum* is an infectious oomycete affecting dogs that develop the cutaneous or gastrointestinal form of pythiosis with a poor prognosis. If left untreated, pythiosis may be fatal. This organism is not a true fungus because its cell wall and cell membrane lack chitin and ergosterol, respectively, requiring specific treatment. Identifying the organism is challenging, as a hematoxylin and eosin (H&E) stain poorly stain the *P. insidiosum* hyphae and cannot be differentiated conclusively from other fungal or fungal-like organisms (such as *Lagenidium* sp.) morphologically. Our study aimed to develop a nested PCR to detect *P. insidiosum* and compare it with the traditional histopathologic detection of hyphae. Formalin-fixed, paraffin-embedded (FFPE) tissue scrolls from 26 dogs with lesions suggesting the *P. insidiosum* infection were assessed histologically, and DNA was extracted from the FFPE tissue sections for nested PCR. Agreement between the histologic stains, (H&E), periodic acid–Schiff (PAS), and/or Grocott methenamine silver (GMS) and the nested PCR occurred in 18/26 cases. Hyphae consistent with *Pythium* sp. were identified via histopathology in 57.7% of the samples, whereas the nested PCR detected *P. insidiosum* in 76.9% of samples, aiding in the sensitivity of the diagnosis of pythiosis in dogs. Using this combination of techniques, we report 20 canine cases of pythiosis over 18 years in Indiana and Kentucky, an unexpectedly high incidence for temperate climatic regions. Using a combination of histopathology evaluation and nested PCR is recommended to aid in the accurate diagnosis of pythiosis.

## 1. Introduction

*Pythium insidiosum* is an oomycte that infects dogs and horses [1] and, infrequently, humans and other vertebrates [2,3,4,5,6,7]. *Pythium* sp. infections typically occur in tropical/subtropical regions; however, a few cases are reported in other areas considered abnormal locations to find this oomycete [4,6,8]. The cutaneous or gastrointestinal forms of pythiosis are the most common types to develop in infected animals, and both forms often have poor prognoses [4] unless diagnosed in an early stage of infection when the complete excision of lesions is possible [4]. This organism is not an actual fungus since its cell wall and cell membrane lack chitin and ergosterol, respectively [4,6,8]. The antifungal treatment of *P. insidiosum* cases is usually not successful due to this unique characteristic of this organism. Therefore, recent studies have suggested that novel drug combinations may be associated with better outcomes [6,9,10,11,12,13]. Morphologically, *P. insidiosum* is typically ~3–10 µm wide, with thick walls, few septa, and close to right-angle branches. The organism identification is challenging, as a hematoxylin and eosin (H&E) stain poorly stains the *P. insidiosum* hyphae. When visualized, it cannot always be definitively distinguished from other fungal or fungal-like organisms (such as *Lagenidium* sp.) based on morphology [3,6,14].

Due to the challenges in differentiating pythiosis infection from fungal and other infections that result in similar inflammatory lesions, molecular diagnosis-based PCR assays were developed to detect the genomic DNA of *P. insidiosum* extracted from cultured organisms or infected tissues [8,15,16,17,18,19]. Ribosomal DNA (rDNA) [20] is the widely used target for detecting fungal DNA [21]. The rDNA of the 18S rRNA gene, inter transcriptional space 1 (ITS1), ITS2, 28SrRNA, and 5.8S rRNA are the most commonly used targets for detecting *P. insidiosum* DNA using universal pan fungal primers [7,22]. The use of pan fungal primers must be followed by sequencing the amplified DNA region to confirm *P. insidiosum*. Specific primers for *P. insidiosum* rDNA detection have been developed to avoid the subsequent DNA sequencing step [15,16,21]. Unfortunately, these primers failed to detect some *P. insidiosum* due to a sequence mismatch with some *P.*
*insidiosum* strains [23,24]. Another study used *P. insidiosum*-specific primers targeting the exo-1,3-b-glucanase gene [25]. In this study, the PCR product was 550 bp, and the authors considered this assay efficient in detecting the *P. insidiosum* strains [3].

Since the therapeutic response is unfortunately poor and patients in late stages succumb to the disease, pythiosis must be accurately diagnosed and treated as early as possible. A faster and more sensitive diagnostic test is a continuously evolving need. Therefore, this study intended to assess the efficiency of using a nested PCR method. In this method, we combine the pan-fungal primers ITS3 and ITS4, followed by species-specific *P. insidiosum* primers to detect the genomic DNA of *P. insidiosum* extracted from formalin-fixed, paraffin-embedded (FFPE) scrolls.

## 2. Materials and Methods

### 2.1. Samples

Twenty-six FFPE samples of different tissues from dogs with lesions suggestive of *P. insidiosum* infection submitted to the Indiana Animal Disease Diagnostic Laboratory (ADDL) between 2001 to 2019 were evaluated (Table 1). The lesions were pyogranulomatous to eosinophilic inflammation in the skin, gastrointestinal tract, and lymph nodes with either a lack of etiologic agent identified or an identification of hyphae consistent morphologically with *Pythium* sp. An H&E stain, GMS stain and PAS stain were also used to aid in identification. All the slides were examined for the presence of hyphae.

### 2.2. DNA Extraction

Four scrolls (20 µm thick) from the FFPE tissue were used for DNA isolation as previously described with some modifications [26]. Briefly, 1 mL of a deparaffinization reagent (d-limonene, QIAGEN^®^ Inc., Valencia, CA, USA) was added to the scrolls to dissolve the paraffin of each sample. The samples were washed with ethanol (1 mL) and digested with an ATL buffer (QIAGEN) and proteinase K (QIAGEN) at 56 °C for 1 h, followed by another 1 h of incubation at 90 °C. The FastPrep-24™ Classic Instrument (MP Biomedicals, Irvine, CA, USA) was used to homogenize the samples for 1 min five times at 5 min intervals to avoid overheating and degradation of the sample. The downstream DNA extraction was performed using the Quick-DNA™ Fungal/Bacterial Miniprep Kit (Zymo Research, Irvine, CA, USA), according to the manufacturer’s instructions. Cultured *Aspergillus fumigatus* was used as the DNA extraction positive control for the kit, and DNase-free water was used as the negative extraction control. The quantity and quality of DNA were assessed by UV spectrophotometry (NanoDrop, ThermoFisher Scientific Waltham, MA USA).

### 2.3. Pan-Fungal Conventional PCR Assay

Conventional PCR was performed using the pan-fungal protocol [27] with the ITS3 forward primer (5’-GCA TCG ATG AAG AAC GCA GC-3’) and ITS4 reverse primer (5’-TCC TCC GCT TAT TGA TAT GC-3’). These pan-fungal primers amplify a DNA fragment of approximately 623 bp of the ITS2 non-coding region, located between the coding regions for 5.8S and the small rRNA gene subunits conserved in virtually all fungal genomes [21,24]. The PCR assay was performed by using the GoTaq^®^ Colorless Master Mix (Promega^®^ Corporation, Madison, WI, USA) and consisted of 12.5 μL of the GoTaq^®^ Colorless Master Mix, 1 μL of each primer ITS3 (10 mM) and ITS4 (10 Μm), 5 μL of nuclease-free water, and 5 μL of the assigned DNA sample. The thermoprofile consisted of enzyme activation at 95 °C for 5 min, followed by 40 cycles of denaturation at 95 °C for 30 s, annealing at 53 °C for 1 min, and extension at 72 °C for 1 min, and then one cycle of final extension at 72 °C for 5 min. Cultured *A.fumigatus* DNA was used as the positive PCR control. Nuclease-free water was used as the negative PCR control.

### 2.4. Nested PCR

The PCR product from the pan-fungal PCR was then used as the template for a second (nested) PCR assay using the newly designed primers NE Fw: 5’-ATG CCT GGA AGT ATG CCT GT-3’ and NE Rev: 5’-TCA CTG CGT TCG AGC ATT AC-3’. These primers were designed to specifically amplify a product size of 191 bp of *P. insidiosum* rDNA. This protocol used 12.5 μL of the GoTaq^®^ Colorless Master Mix, 1 μL of each primer NE Fw (10 Μm) and NE Rv (10 Μm), 9.5 μL of nuclease-free water, and 1 μL of the assigned sample. The thermoprofile consisted of enzyme activation at 95 °C for 5 min, followed by 40 cycles of denaturation at 95 °C for 30 s, annealing at 56 °C for 1 min, and extension at 72 °C for 1 min. One cycle of final extension at 72 °C for 5 min followed. *P. insidiosum* DNA isolated from a known positive clinical sample was used as the positive PCR control and nuclease-free water as the negative control. After amplification, the PCR products and molecular weight markers (GeneRuler 1 kb Plus DNA Ladder, ThermoFisher Scientific) were loaded into a 1.5% agarose gel with 0.5 μg/mL ethidium bromide and separated by electrophoresis for 1 h at 100 V. Visualization and documentation was achieved by exposing the gel to 312 nm UV light (G: BOX XT4: Chemiluminescence and Fluorescence Imaging System, Syngene, Frederick, MD, USA).

### 2.5. Conventional PCR for the GAPDH Gene

Negative nested PCR samples were subject to a conventional PCR targeting glyceraldehyde-3-phosphate dehydrogenase (GAPDH), the housekeeping gene, to verify the presence of amplifiable DNA [27].

### 2.6. Sequencing

Amplicons with the expected sizes (623 bp or 191 bp) were excised from the gel, and DNA was purified using a Zymoclean™ Gel DNA Recovery Kit (Zymo Research, Irvine, CA, USA). Sequencing was performed by the Sanger method [28] at the Genomics Core Facility at Purdue University.

### 2.7. Cloning of the PCR Amplicons

The NE primers were used to amplify the 191 bp amplicon using a Phusion Hot Start II DNA Polymerase (ThermoFisher Scientific, Waltham, MA, USA). Gel electrophoresis was used to purify the PCR product, and bands with the expected size were purified (Invitrogen™ PureLink™ Quick Gel Extraction Kit #K210012, Waltham, MA, USA). A Fast DNA End Repair Kit (ThermoFisher Scientific, Waltham, MA, USA) was used to add the phosphate groups required for ligation to the amplicons. The EcoRV restriction enzyme was used to linearize the pCDNA3.1 plasmid, followed by using the quick CIP dephosphorylation kit (New England Biolabs, Ipswich, MA, USA). The phosphorylated amplified PCR product was ligated to the linearized pCDNA3.1 plasmid using the rapid ligation kit (ThermoFisher Scientific) to create the pCDNA3.1-Pythium-ITS2 366-556 plasmid (Figure 1). The ligation mixture was used to transform Stellar™ Competent Cells (Takara Bio USA, Mountain View, CA, USA). The correct clone was identified, and the cloned plasmid was purified from a 200 mL bacterial culture using a Maxi Fast-Ion Plasmid Kit (IBI Scientific, Road Dubuque, IA, USA). The DNA yield was 1.28 mg, and the 260/280 ratio was 1.85.

### 2.8. Determination of PCR Sensitivity

The minimum copy number needed for detection was determined using the pCDNA3.1-Pythium-ITS2 366-556 plasmid diluted in nuclease-free water to 20 ng/µL. The genome copy number was calculated for pCDNA3.1-Pythium-ITS2 366-556 using the plasmid size (5206 bp) and the DNA concentration using the DNA copy number and dilution calculator [29] and the calculator for determining the template of the copy number [30]. Ten-fold dilutions of the plasmid were then used to calculate the minimum detection limit of the copies by the nested PCR assay.

## 3. Results

The histopathology and nested PCR were in agreement in 18/26 (69.2%) of the cases: 15/26 (57.7%) of the cases had hyphae identified histologically via the H&E, GMS, and/or PAS stains (Figure 2) and were also positive for *P. insidiosum* by the nested PCR (Figure 3 and Appendix A), and 3/26 (11.5%, cases 6, 10, and 18) of the cases were negative histologically for hyphae identification and by the nested PCR and, therefore, considered negative for the *P. insidiosum* infection. Interestingly, case 18 was positive for the pan-fungal PCR (ITS3 and ITS4 primers), and the sequencing revealed the presence of *A. fumigatus*.

Disagreement between the histopathology and nested PCR occurred in 8/26 cases: in 5/26 (19.2%, cases 14, 15, 20, 22, and 24), hyphae were not identified histologically via the H &E, GMS, and/or PAS stains but were positive by the nested PCR. On the other hand, 3/26 cases (11.5%, cases 8, 9, and 12) were positive by the histopathology but negative by the nested PCR. A summary of the case results is shown in Table 1.

Seven PCR products from the nested PCR were sent for sequencing to verify the specificity of the nested PCR assay: the seven sequences were 100% identical to the *P. insidiosum* sequences (Sequence IDs: GQ260125.1, GU137331.1, GU137328.1, and EF016914.1) deposited in the Genbank^®^ (Bethesda, MD, USA) nucleotide database [31]. A representative sequence was then submitted to the Genbank^®^ database under accession number: OM282097.

The cloning of pCDNA3.1-Pythium-ITS2 366-556 was successful, and the large-scale plasmid preparation provided a DNA yield of 1.28 mg with a 260/280 ratio of 1.85. The minimum copy number detection using the NE primers for the nested PCR was 0.05, corresponding to five copies in 100 µL (Figure 4 and Appendix A).

## 4. Discussion

*P. insidiosum* is a fungal-like organism that can infect dogs, mostly in tropical and subtropical regions [32]. The clinicopathological findings of this infection are characterized by pyogranulomatous inflammation in the skin or the gastrointestinal tract. Due to the complexities associated with identifying or isolating *P. insidiosum*, a conclusive diagnosis is hard to attain. Diagnosing *P. insidiosum* by morphology alone is presumptive given the similarities with other phycomycotic or zygomycotic organisms, usually resulting in erroneous diagnosis and delayed treatment [32]. Similarly, immunohistochemistry using anti-*P. insidiosum* antibodies may cross-react with other related fungal species, preventing a definitive diagnosis [33]. The molecular diagnosis is of great importance in identifying the organisms’ genome, especially if other diagnostic tools are inconclusive. Thus, we have developed a novel nested PCR assay to help identify *P. insidiosum* in FFPE sections from different organs from dogs suspected of pythiosis based on the presence of pyogranulomatous to eosinophilic inflammation and a consistent history or location of lesions.

When choosing the clinical cases for this study, we were interested in cases with granulomatous to eosinophilic dermatitis, enteritis, colitis, mesenteritis, pancreatitis, and lymphadenitis, with either visualized hyphal structures in the sections consistent with *P. insidiosum* or the suspicion of pythiosis based on the location and nature of the inflammatory reaction. Most of the cases positive by H& E [34] staining (13/26) were also positive with either GMS [35] or PAS staining; however, a few cases were negative by H&E staining but positive for PAS (case 4) or GMS (cases 7, 9, 26) staining. These results are in agreement with Mendoza and collaborators [34], who described the challenges in detecting the *P. insidiosum* hyphae using an H &E stain but increased sensitivity using a GMS stain, showing the advantage of using special stains to help identify fungal organisms [34]. Regarding the PAS stain, Mittal and collaborators [36] reported the variability or poor staining of *P. insidiosum* hyphae due to the presence of cellulose in their cell walls. Thus, the PAS stain requires a longer exposure time and a higher concentration of periodic acid to generate the magenta color [36,37]. In the same study, the authors proposed a new staining technique for diagnosing *P. insidiosum* using potassium iodide–sulfuric acid (IKI-H_2_SO_4_). Although more sensitive and cost-effective than the PAS stain, using the IKI-H_2_SO_4_ stain entails caution in handling highly concentrated sulfuric acid (65%) [36]. Also, this technique requires well-trained professionals to operate a florescent microscope that is not always used or accessed in many clinics [38].

Several researchers have used molecular diagnostic tools for diagnosing P. insidiosum, including a pan-fungal primers’ PCR followed by sequencing [19,39,40]. However, this method requires highly purified samples and is expensive for regular clinical diagnosis [41]. Some studies have used *P. insidiosum*-specific primers, but they either used a small sample size (*n* = 6) [15] or fresh tissue or cultured organisms as the starting material [23,42]. Our study used a nested PCR assay to detect the *P. insidiosum* DNA from the FFPE blocks from different organs with pyogranulomatous to eosinophilic inflammation and a consistent history or location of pythiosis lesions. Our novel assay has also proven sensitive, with a detection limit of five copies in 100 µL. We have found that 20 samples out of the 26 are positive for *P. insidiosum* (20/26), with a nearly 77% positive rate in the suspected samples. Five of these samples were negative in the hyphae staining and positive nested PCR for *P. insidiosum* 19.2% (5/26), corroborating the limitations of the histopathological diagnosis of pythiosis. Another study has shown that the PCR successfully detected *P. insidiosum* in 6 of 21 (28.57%) clinical samples that failed to grow in culture and also emphasized the limitation of the microbiological diagnosis of pythiosis [42]. The addition of a nested PCR assay increased the sensitivity for *P. insidiosum* from 57.7% (15/26) to 76.9% (20/26 samples) positivity. Thus, the specific *Pythium* sp. nested PCR is more sensitive than histopathology alone and should be added to the diagnostic profile in cases where *P. insidiosum* is strongly suspected, but organisms are not seen using H&E, GMS, or PAS staining. This combinatory approach could more accurately direct the therapeutic approach.

A disagreement between histopathology and nested PCR occurred in 8/26 of the cases. Five cases (14, 15, 20, 22, and 24) were positive by the nested PCR, but hyphae were not identified histologically via H&E, GMS, and/or PAS staining. *P. insidiosum* is notoriously tricky to identify on H&E staining because it is non-staining and appears as a clearing in the shape of hyphae surrounded by an eosinophilic rim [2]. These organisms either stain poorly or do not stain with the PAS stain due to the lack of chitin within the cell wall; therefore, the lack of PAS staining cannot rule out infection with this organism. The GMS stain should stain these organisms well; however, it is possible that hyphae were not present in the examined histologic cut. Since PCR is generally more sensitive and able to detect the presence of DNA whether or not intact fungal hyphae are present, the lack of the identification of hyphae histologically is likely due to a combination of low numbers of organisms, difficulty visualizing the hyphae, or the absence of hyphae in the tissue cuts examined histologically. The characteristics of the lesions combined with a positive nested PCR and sequencing provide confidence for diagnosing pythiosis in all five of these cases.

The three cases with histopathology positive and nested PCR negative (8, 9, 12) had hyphae consistent with *P. insidiosum* identified on the histopathology, but the hyphae were fragmented. In cases 8 and 12, the hyphae were 7–10 µm wide, had roughly parallel walls, and with rare septa and rare right-angle branching, and in case 9, hyphae were 5–10 µm wide with few septa (the fragmentation prevented further characterization). It is possible that these hyphae are not *P. insidiosum,* and other etiologies should be considered. *Lagenidium* sp. has almost identical hyphae (6–20 µm wide, rare septa, and right-angle branching) and is, therefore, a top differential diagnosis in this case [4]. Additionally, zygomycosis, such as *Basidiobolus* sp. (5–20 µm wide, rare septa, and non-dichotomous branching) and *Conidiobolus* sp. (5–13 µm wide, rare septa, and non-dichotomous branching), can also appear similar to [4] and should be ruled out.

The occurrence of 20 cases of canine pythiosis in 18 years is higher than expected in Indiana and Kentucky, USA (the northern temperate climate zone), where the study was conducted, compared to four cases reported in a period of nine years in an epidemiological study in the same area [43]. A few clinical cases of pythiosis in dogs are reported in non-tropical regions, such as two cases of cutaneous pythiosis in a dog from Wisconsin, USA [44], ten canine cases reported in California, USA [45], and dogs in Arizona, USA [4,46,47]. The presence of *P. insidiosum* in temperate climate areas raises awareness for broader geographic distribution. It highlights the importance of recognizing the clinical and histopathological features of the disease, as well as the available diagnostic tools to identify this organism correctly. A study investigating the epidemiology of this disease in the midwestern region of the USA is ongoing.

As previously mentioned, *P. insidiosum* is exceptionally challenging to visualize histopathologically, even with special stains, so it is always possible to miss the etiologic agent using histopathology alone [4]. The novel nested PCR reported herein may be used as an ancillary tool to help confirm *P. insidiosum* from biopsy specimens providing a fast and accurate diagnosis paramount to initiating the most appropriate treatment. Given that *P. insidiosum* is an oomycete lacking ergosterol in the cell membrane and chitin in the cell wall [5,6], typical fungicides are ineffective because they target either ergosterol synthesis (i.e., the azoles) or alter membrane permeability (i.e., amphotericin B) [46], emphasizing the importance of an accurate diagnosis. Other medications, such as caspofungin (which inhibits B-glucan synthesis) and mefenoxam (which inhibits RNA polymerase), are promising therapies but have not been thoroughly evaluated to date [11]. Therapeutic options include surgical excision and immunotherapy, but the overall therapeutic response is low, and most patients do not survive [46]. Regardless of the treatment chosen, pythiosis must be correctly diagnosed and therapy started immediately. The developed *P. insidiosum*-specific nested PCR allows for a rapid and more sensitive and precise diagnosis that enables clinicians to initiate the correct treatment earlier and give the dog a better chance of surviving.

## 5. Conclusions

Infection caused by *P. insidiosum* is life-threatening to dogs if left untreated. Pythiosis may also affect humans and other animals, such as horses and cattle, and is more common in humid tropical and subtropical climatic zones. An accurate pythiosis diagnosis is essential for successful treatment, while a histopathological examination using multiple stains to detect the organism’s hyphae is not conclusive. Herein, we describe a novel nested PCR assay to detect the *P. insidiosum* infection from FFPE samples. Using this sensitive and accurate molecular diagnostic technique enabled us to record 20 cases over 18 years in Indiana and Kentucky, an unexpectedly high incidence for such temperate climatic regions. Our study raises awareness of the wider geographic distribution of this organism in addition to underscoring the significance of incorporating the nested PCR assay with the histologic assessment to improve the diagnostic sensitivity of pythiosis in dogs.

## Figures and Tables

**Figure 1 vetsci-09-00444-f001:**
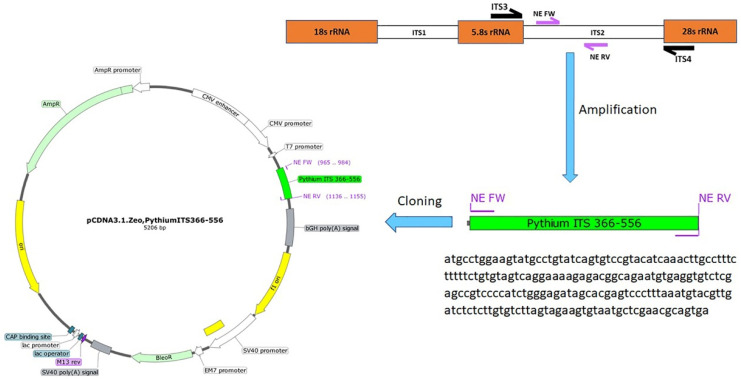
Cloning steps of the pCDNA3.1-Pythium-ITS2 366-556 plasmid were developed to estimate the detection limit of the nested PCR assay, including a diagram showing the location of the primers used for the development of a nested PCR. The ITS2 and ITS4 primers have been previously described. The NE Fw and NE Rv primers were designed in this study. The size of the products is based on the *P. Insidiosum* sequence (GenBank ID: GQ260125.1).

**Figure 2 vetsci-09-00444-f002:**
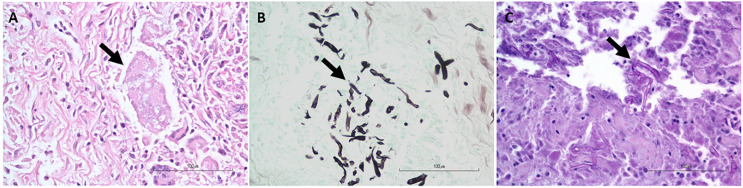
Representative photomicrographs for the histopathologic identification of hyphae morphologically consistent with *Pythium* sp. (arrows). (**A**). H&E stain; (**B**). GMS stain; and (**C**). PAS stain.

**Figure 3 vetsci-09-00444-f003:**
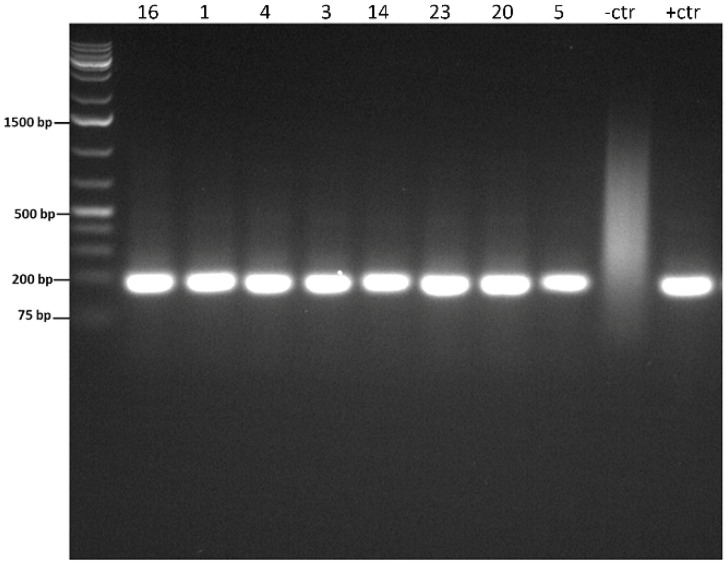
Representative picture of the nested PCR gel’s electrophoresis, showing positive bands with approximately 191 bp correspondent to *P. insidiosum*. The negative control was nuclease-free water, and the positive control was *P. insidiosum* DNA. DNA Ladder (L), base pair size markers; numbers refer to clinical cases; negative control (-ctr); positive control (+ctr).

**Figure 4 vetsci-09-00444-f004:**
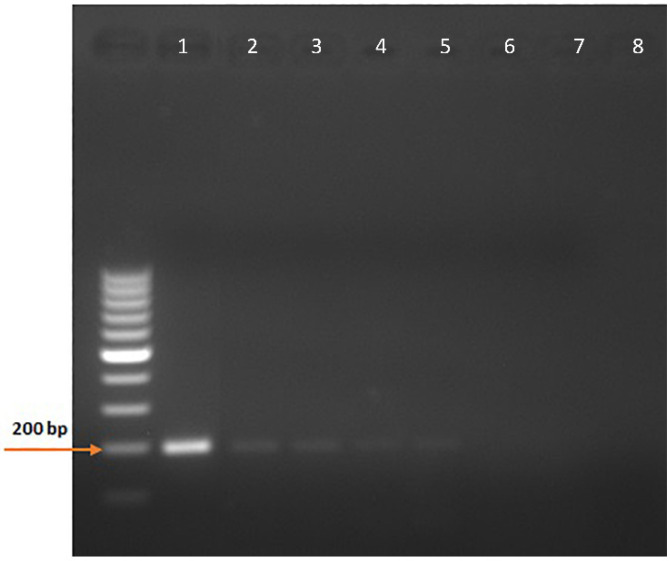
Minimum copy number detection for the nested PCR assay. Ten-fold dilutions of the pCDNA3.1-Pythium-ITS2 366-556 plasmid were used to determine the sensitivity of the nested PCR assay detecting the ITS2 366-556 sequences of *P. insidiosum* rDNA. 1, 5 × 10^2^ copies; 2, 5 × 10^1^ copies; 3, 5 × 10^0^ copies; 4, 5 × 10^−1^ copies; 5, 5 × 10^−2^ copies; 6, 5 × 10^−3^ copies; 7, 5 × 10^−4^ copies; 8, No template.

**Table 1 vetsci-09-00444-t001:** Summary of the results of each case selected and comparisons between the microscopic observation of hyphae in the H&E, GMS, and PAS stains to the nested PCR assay developed in this study.

Case Number	Age (Years)	Sex	Organ (s) Affected	H & E	GMS	PAS	Nested PCR
**1**	1.5	F	Pancreas	+	-	+	+
**2**	2	F	Stomach	+	+	+	+
**3**	1	M	Duodenum	+	-	+	+
**4**	1.5	M	Stomach	-	-	+	+
**5**	2	F	Stomach, lymph nodes	+	+	+	+
**6**	2.5	FS	Liver, mesenteric nodule	-	-	-	-
**7**	0.10	M	Mesenteric mass	-	+	-	+
**8**	1	F	Small intestine	+	+	+	-
**9**	1	M	Small intestine	-	+	-	-
**10**	3	M	Mesentery	-	-	-	-
**11**	2	MN	Large Intestine	N/A	+	-	+
**12**	2	MN	Small intestine, colon, rectum	+	+	+	-
**13**	3	M	Mesenteric lymph node	+	+	+	+
**14**	4	MN	Proximal duodenum, liver	-	-	-	+
**15**	8.5	F	Small Intestine, pancreas	-	-	-	+
**16**	2	M	Pylorus, small intestine, mesenteric lymph node	+	+	+	+
**17**	3.5	FS	Jejunum	+	+	+	+
**18**	10	FS	Ileum	-	-	-	- *
**19**	4	MN	Stomach, duodenum, lung, lymph node	+	+	+	+
**20**	5.5	MN	Colon, mesenteric root, mesenteric lymph nodes	-	-	-	+
**21**	2	MN	Colon, ileocecal junction, mesenteric lymph node	+	+	+	+
**22**	6	M	Intestine, omentum, lymph node	-	-	-	+
**23**	7	FS	Left ventral thoracic lesion	+	+	+	+
**24**	1	FS	Ileum, mesenteric lymph node	-	-	-	+
**25**	0.5	U	Skin	+	+	+	+
**26**	0.10	FS	Proximal jejunum, liver abdominal mass, intestinal mass, mesenteric lymph node	-	+	-	+

* Identified as *A. fumigatus* by sequencing the product of the first PCR assay with primers ITS2 and ITS4. N/A: not available.

## Data Availability

The data are contained within the article or Appendix A. A representative sequence of a *Pythium insidiosum* 5.8S ribosomal RNA gene and internal transcribed spacer 2 was submitted to the Genbank^®^ database under accession number: OM282097.

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
