# Peer review of "Nested PCR Detection of Pythium sp. from Formalin-Fixed, Paraffin-Embedded Canine Tissue Sections"

_vetsci, 2022, doi:10.3390/vetsci9080444_

Round 1
Reviewer 1 Report
In whole, manuscript is well structuraly organized, all of the sections is well described, quality of presentation is very good and supported by adequate discussion. In conclusion summarized all of the relevant facts.
This manuscript discussed the molecular diagnosis-based PCR assays detect P. insidiosum genomic DNA extracted from formalin-fixed, paraffin-embedded canine tissue sections. Due to the difficulty of differentiating pythiosis infection from fungal and other inections that result in a similar inflammatory lesion, molecular diagnosis-based PCR says were widely applied for detecting fungal DNA. It is of practical significance to detect/study the P. insidiosum. However, in terms of the current manuscript quality, some necessary details and discussions are lacking. The authors did not provide enough data in section discussion to make this section systematic or interesting.
Orginality of the topic is mainly related to the developet methodology which will be of importance for the improving diagnosis and therapeutic of diseases. Proposed methodology for diagnosis of pathogenic fungi is more accurately, then methodology based on morphology.
Manuscript is well organized in logical order, all of the important section and subsection with relevant data is provided, tables, figures etc. Therefore, manuscript is easy to read and understand. As before, some necessary details and discussions are lacking, thus the discussion is poor and must be imporoved.
Conclusion is completely missing and must be drawn.
In overal, manuscript must be revised, major revision.

Author Response
Thank you so much for your comments. We addressed all your comments, and the corrections are provided in the manuscript on the lines indicated below.
- Line 94 " insidiosum" italicized
- Line 99 "Pythium sp." italicized
- Line 182 "ng/µl" Change to "ng/µL"
- Line 191 " insidiosum" italicized
- Line 224 "µl" Change to "µL"
We added a conclusion section to the manuscript.
- Lines 299-305:
"Pythiosis is more common in humid tropical and subtropical climatic zones. Herein, we are reporting 20 cases over 18 years in Indiana and Kentucky, an unexpectedly high incidence for such temperate climatic regions. Our study raises awareness of the wider geographic distribution of this organism in addition to underscoring the significance of incorporating the nested-PCR assay with the histologic assessment to improve the diagnostic sensitivity of pythiosis in dogs."
Reviewer 2 Report
The manuscript is well written and clearly presented, from the introduction to the results. The subject is interesting and of considerable importance for accurate diagnosis and rapid implementation of treatment. However, I missed some information, which I expose below:
Was the DNA not quantified after extraction? If yes, describe the protocol
Because the hybridization temperature used was 53-56°, since higher temperatures can increase the specificity of the reaction.
Was there no statistical analysis? If yes, describe.
Author Response
We appreciate your comments. We addressed all your comments, and the corrections are provided in the manuscript on the lines indicated below.
Comments to the Author
- Was the DNA not quantified after extraction? If yes, describe the protocol.
Response: Yes, we used UV spectrophotometry (Nanodrop) to evaluate the quantity and the quality of the extracted DNA.
Lines 116-117:
"The quantity and quality of DNA were assessed by UV spectrophotometry (NanoDrop, ThermoFisher Scientific, MA USA)."
- Because the hybridization temperature used was 53-56°, since higher temperatures can increase the specificity of the reaction.
Response: On the first PCR round, we ran a pan-fungal PCR using an annealing temperature of 53°C; afterward, we ran the nested-PCR with an annealing temperature of 56°C. The annealing temperatures were based on the primers used. To confirm the identity of the amplified products, representative samples were sent for sequencing, and the identity was verified as described in the results (lines 216-221).
- Was there no statistical analysis? If yes, describe.
Response: We did not perform statistical analysis since we only detected the presence of P. insidiosum. Instead, we used basic percentage calculation as described on lines 189-193.
Reviewer 3 Report
In this manuscript the authors have demonstrated how a nested-PCR method is more efficient than histopathological staining methods in detecting the oomycete plant pathogen Pythium insidiosum in FFPE sections of tissues from dogs suspected of pythiosis. The PCR method allows a rapid and more sensitive and precise diagnosis of pythiosis, which enables appropriate treatment to be administered earlier and gives the dog a better chance of survival.
The manuscript is generally well written, and the results are presented clearly. I suggest that the manuscript text subject to substantial corrections and changes, as follows:
Line 16. Define “H&E”
Line 18. Define “FFPE”
Line 21. Define “PAS” and “GMS”
Line 35. Is “guarded” the intended word to use here?
Line 36. Insert a space after “possible”
Line 70. Italicise “P. insidiosum”
Lines 74-75. Change “silver stain (GMS) and periodic acid–Schiff stain (PAS) stained slides were also obtained” to “silver (GMS) stain and periodic acid–Schiff (PAS) stain were also used to stain slides”
Lines 75-76. Change “The authors reviewed all the slides” to “All slides were examined”
Line 83. Insert “a” after “using”
Line 86. Change “the” to “a”
Line 88. Italicise “Aspergillus fumigates”
Line 96. Remove “the”
Line 102. Change “Aspergillus fumigates” to “A. fumigates” and italicise
Line 108. Italicise “P. insidiosum”
Line 113. Italicise “P. insidiosum”
Lines 115-116. Change “and a molecular weight marker” to “and molecular weight markers”
Line 127. Insert “a” after “using”
Line 134. Change “The” to “A”
Line 136. Insert “restriction” before “enzyme” and “the quick” to “by use of the quick”
Line 139. Insert “the” after “create”
Line 142. Insert “a” before “Maxi”
Line 151. Change “Pythium Insidiosum” to “P. insidiosum”
Line 153. Insert “the” after “using”
Line 157. Insert a space after “[29]”
Line 163. Italicise “P. insidiosum”
Line 165. Italicise “P. insidiosum”
Line 167. Change “Aspergillus fumigates” to “A. fumigates” and italicise
Line 168. Change “of histopathologic” to “for histopathologic”
Line 173. Change “Pythium insidiosum” to “P. insidiosum” and italicise, and change “consists of” to “was”
Line 174. Change “is Pythium insidiosum” to “was P. insidiosum”
What is meant by “L.”?
Change “molecular marker” to “base pair size markers”
If S16-S23 are clinical cases what do the other samples refer to?
Lines 182-183. Change “hematoxylin-eosin stain (H&E), Grocott's methenamine silver stain (GMS), and periodic acid–Schiff stain (PAS)” to “H&E, GMS, and PAS stains”
Line 187. Change “sent to” to “sent for”
Line 188. Italicise “P. insidiosum”
Line 193. Change “provided 1.28 mg DNA yield with the” to “provided a DNA yield of 1.28 mg with a”
Line 197. Change “nester” to “nested”
Line 198. Change “Pythium insidiosum” to “P. insidiosum”
Line 202. Change “Pythium insidiosum” to “P. insidiosum” and italicise
Line 210. Italicise “P. insidiosum”
Line 215. Change “the authors” to “we”
Lines 217-218. Italicise “P. insidiosum”
Line 219. Insert “staining” after “H&E”
Line 220. Insert staining after “PAS”
Line 222. Italicise “P. insidiosum”
Line 225. Italicise “P. insidiosum”
Line 226. Insert “staining” after “PAS”
Line 230. Insert “staining” after “PAS” and italicise “P. insidiosum”
Line 231. Change “they are” to “it is” and “appear” to “appears”
Line 243. Italicise “P. insidiosum”
Line 247. Italicise “P. insidiosum”
Line 254. Change “Pythium insidiosum” to “P. insidiosum” and italicise
Line 256. Specify what assay
Lines 256-257. Italicise “P. insidiosum”
Lines 258-259. Italicise “P. insidiosum”
Lines 265-266. Italicise “P. insidiosum”
Line 285. Italicise “P. insidiosum”
Lines 287-288. Italicise “P. insidiosum”
Line 293. Italicise “P. insidiosum”
Line 301. Italicise “P. insidiosum”
Line 306. Italicise “P. insidiosum”
Line 309. Italicise “P. insidiosum”
Line 311. Italicise “P. insidiosum”
Line 315. Italicise “P. insidiosum”
Line 320. Italicise “P. insidiosum”
Line 323. Italicise “P. insidiosum”
Line 328. Italicise “P. insidiosum”
Line 334. Italicise “P. insidiosum”
Line 350. Italicise “P. insidiosum”
Line 354. Italicise “P. insidiosum”
Author Response
Thank you so much for your thoughtful comments and helpful edits. All the substantial corrections and changes are resolved and can be found in the manuscript.
- Line 16. Define "H&E"
Response: We defined it in line 28.
- Line 18. Define "FFPE"
Response: We defined it in line 31.
- Line 21. Define "PAS" and "GMS"
Response: We defined it in lines 35-36.
- Line 35. Is "guarded" the intended word to use here?
Response: Yes, we made it clearer "The cutaneous or gastrointestinal forms of pythiosis are the most common types to develop in infected animals, and both forms often have poor prognoses" in lines 51-53.
- Line 36. Insert a space after "possible"
Response: We did it in line 56.
- Line 70. Italicise "P. insidiosum"
Response: We did in line 94.
- Lines 74-75. Change "silver stain (GMS) and periodic acid-Schiff stain (PAS) stained slides were also obtained" to "silver (GMS) stain and periodic acid-Schiff (PAS) stain were also used to stain slides"
Response: We did in lines 99-100.
- Lines 75-76. Change "The authors reviewed all the slides" to "All slides were examined"
Response: We did in line 101.
- Line 83. Insert "a" after "using"
Response: We did in line 84.
- Line 86. Change "the" to "a"
Response: We changed the sentence to "The FastPrep-24™ Classic Instrument (MP Biomedicals, Irvine, CA, USA) was used to homogenize the samples for 1 minute five times with 5 minutes intervals to avoid overheating and degradation of the sample." in lines 108-111.
- Line 88. Italicise "Aspergillus fumigates"
Response: We did in line 114.
- Line 96. Remove "the"
Response: We did in line 123.
- Line 102. Change "Aspergillus fumigates" to "A. fumigates" and italicise
Response: We did in line 130.
- Line 108. Italicise "P. insidiosum"
Response: We did in line 136.
- Line 113. Italicise "P. insidiosum"
Response: We did in line 114.
- Lines 115-116. Change "and a molecular weight marker" to "and molecular weight markers"
Response: We did in lines 144.
- Line 127. Insert "a" after "using"
Response: We did in line 155.
- Line 134. Change 'The" to "A"
Response: We did in line 162.
- Line 136. Insert "restriction" before "enzyme" and "the quick" to "by use of the quick"
Response: We did in line 164.
- Line 139. Insert "the" after "create"
Response: We did in line 168.
- Line 142. Insert "a" before "Maxi"
Response: We did in line 171.
- Line 151. Change "Pythium lnsidiosum" to "P. insidiosum"
Response: We did in line 179.
- Line 153. Insert "the" after "using"
Response: We did in line 181.
- Line 157. Insert a space after "[29]"
Response: We did in line 185.
- Line 163. Italicise "P. insidiosum"
Response: We did in line 191.
- Line 165. Italicise "P. insidiosum"
Response: We did in line 193.
- Line 167. Change "Aspergillus fumigates" to "A. fumigates" and italicize
Response: We did in line 195.
- Line 168. Change "of histopathologic" to "for histopathologic"
Response: We did in line 196.
- Line 173. Change "Pythium insidiosum" to "P. insidiosum" and italicise, and change "consists of' to "was"
Response: We did in line 201.
- Line 174. Change "is Pythium insidiosum" to "was P. insidiosum"
Response: We did in line 202.
- What is meant by "L."?
Response: We meant DNA Ladder. We added "DNA Ladder(L)" in line 202.
- Change "molecular marker" to "base pair size markers"
Response: We did in lines 202-203.
- If S16-S23 are clinical cases what do the other samples refer to?
Response: We deleted "S16-S23", and added "numbers refer to clinical cases" to be clearer in line 203.
- Lines 182-183. Change "hematoxylin-eosin stain (H&E), Grocott's methenamine silver stain (GMS), and periodic acid-Schiff stain (PAS)" to "H&E, GMS, and PAS stains"
Response: We did in lines 211-212.
- Line 187. Change "sent to" to "sent for"
Response: We did in line 216.
- Line 188. Italicise "P. insidiosum"
Response: We did in line 218.
- Line 193. Change "provided 1.28 mg DNA yield with the" to "provided a DNA yield of 1.28 mg with a"
Response: We did in line 223.
- Line 197. Change "nester" to "nested"
Response: We did in line 227.
- Line 198. Change "Pythium insidiosum" to "P. insidiosum"
Response: We did in line 228.
- Line 202. Change "Pythium insidiosum" to "P. insidiosum" and italicize
Response: We did in line 232.
- Line 210. Italicise "P. insidiosum"
Response: We did in line 240.
- Line 215. Change "the authors" to "we"
Response: We did in line 245.
- Lines 217-218. Italicise "P. insidiosum"
Response: We did in lines 247-248.
- Line 219. Insert "staining" after "H&E"
Response: We did in line 249-250.
- Line 220. Insert staining after "PAS"
Response: We did in line 256.
- Line 222. Italicise "P. insidiosum"
- Response: "P. insidiosum" is not written in the original line 222.
- Line 225. Italicise "P. insidiosum"
Response: We did in line 250.
- Line 226. Insert "staining" after "PAS"
Response: We did in line 250.
- Line 230. Insert "staining" after "PAS" and italicise "P. insidiosum"
Response: We did in line 256.
- Line 231. Change "they are" to "it is" and "appear" to "appear"
Response: We did in line 261.
- Line 243. Italicise "P. insidiosum"->
Response: We did in line 273.
- Line 247. Italicise "P. insidiosum"
- Response: "Pythium insidiosum" was abbreviated and italicized as " insidiosum"
in all the subsequent lines in the manuscript.
- Line 256. Specify what assay
Response: We added "the minimum copy number detection assay" instead of the assay in line 286.
Round 2
Reviewer 1 Report
My opinion are as follows.
An abstract briefly and precisely describes what a paper is about and what readers can anticipate finding from it.
In section Introduction, authors opened a dialogue about the problem and what gaps in diagnosis of pythiosis their aim to bridge and improve.
In section Discussion author did not compare obtained results with similar research, and therefore this section must be improved.
A conclusion must be more extensive and encompassing and must be improved in a way that stresses the significance of this study.
